# Antioxidants and Atherosclerosis: Mechanistic Aspects

**DOI:** 10.3390/biom9080301

**Published:** 2019-07-25

**Authors:** Khojasteh Malekmohammad, Robert D. E. Sewell, Mahmoud Rafieian-Kopaei

**Affiliations:** 1Department of Animal Sciences, Faculty of Basic Sciences, Shahrekord University, Shahrekord 8818634141, Iran; 2Cardiff School of Pharmacy and Pharmaceutical Sciences, Cardiff University, Cardiff CF10 3NB, UK; 3Medical Plants Research Center, Basic Health Sciences Institute, Shahrekord University of Medical Sciences, Shahrekord 8813833435, Iran

**Keywords:** atherosclerosis, oxidative stress, reactive oxygen species, LDL oxidation, antioxidant, medicinal plants

## Abstract

Atherosclerosis is a chronic inflammatory disease which is a major cause of coronary heart disease and stroke in humans. It is characterized by intimal plaques and cholesterol accumulation in arterial walls. The side effects of currently prescribed synthetic drugs and their high cost in the treatment of atherosclerosis has prompted the use of alternative herbal medicines, dietary supplements, and antioxidants associated with fewer adverse effects for the treatment of atherosclerosis. This article aims to present the activity mechanisms of antioxidants on atherosclerosis along with a review of the most prevalent medicinal plants employed against this multifactorial disease. The wide-ranging information in this review article was obtained from scientific databases including PubMed, Web of Science, Scopus, Science Direct and Google Scholar. Natural and synthetic antioxidants have a crucial role in the prevention and treatment of atherosclerosis through different mechanisms. These include: The inhibition of low density lipoprotein (LDL) oxidation, the reduction of reactive oxygen species (ROS) generation, the inhibition of cytokine secretion, the prevention of atherosclerotic plaque formation and platelet aggregation, the preclusion of mononuclear cell infiltration, the improvement of endothelial dysfunction and vasodilation, the augmentation of nitric oxide (NO) bioavailability, the modulation of the expression of adhesion molecules such as vascular cell adhesion molecule-1 (VCAM-1) and intercellular adhesion molecule-1 (ICAM-1) on endothelial cells, and the suppression of foam cell formation.

## 1. Introduction

Atherosclerosis is a chronic inflammatory disease which is a major cause of coronary heart disease and stroke in humans [1]. It is characterized by intimal plaques and cholesterol accumulation in the arterial walls [2]. The term atherosclerosis, from its Greek origin, has two parts, namely Atherosis—characterized by fat accumulation along with macrophages—and the term sclerosis, characterized by a fibrotic layer including smooth muscle cells, connective tissue, and leukocytes [3].

Oxidative stress, which is exemplified by the overproduction of reactive oxygen species (ROS) and oxidized low-density lipoprotein (Ox-LDL), has a pivotal role in the progression of cardiovascular disease linked to atherosclerosis. An imbalance between radical production (reactive oxygen and/or nitrogen species formation) and radical scavenging systems (the antioxidant defense system) is the main cause of oxidative stress [4]. In addition, important risk factors including hypertension, diabetes mellitus, insulin resistance, obesity hypercholesterolemia, dyslipidemia, a high level of C-reactive protein (CRP), stress, alcohol consumption, smoking, an immunological disorder, vascular wall inflammation, a genetic predisposition, and bacterial infection have all been implicated in the development of atherosclerosis [5,6]. 

The side effects of currently prescribed synthetic drugs and their high cost in the treatment of atherosclerosis has prompted the use of alternative herbal medicines, dietary supplements and antioxidants associated with fewer adverse effects in atherosclerosis treatment [7,8]. Largely due to the fact that ROS and the production of oxidized LDL are principal causes of atherosclerotic progression, the utilization of antioxidants may well represent a rational therapeutic strategy to prevent the development of the condition [8].

Despite the availability of extensive literature concerning antioxidants and atherosclerosis, much less attention has been devoted to the mechanistic aspects of antioxidant activity on this health problem. Consequently, there is a need for a comprehensive mechanistic literature review on antioxidants in the prevention and treatment of atherosclerosis. In addition, other than such a mechanistic analysis, the most notable medicinal plants with antioxidant activity against this multifaceted disease are disclosed. 

## 2. Material and Method

The wide-ranging information in this review article was searched from scientific databases including PubMed, Web of Science, Scopus, Science Direct and Google Scholar. The main key words used in this study were: “Atherosclerosis” and “LDL oxidation,” “oxidative stress,” “reactive oxygen species,” “antioxidant,” and “medicinal plants.”

## 3. Results

### 3.1. Oxidative Stress and the Atherosclerotic Process

In general, pro-oxidant substances may be designated as either free radical species or non-radical species that mediate peroxidation. The two major sub-groups are ROS and reactive nitrogen species (RNS). The principal reactive free radical species include superoxide, hydroxyl, hydroperoxyl, peroxyl, nitric oxide and nitrogen dioxide [9,10], whilst the main reactive non-radical species group consist of hydrogen peroxide, peroxynitrite and nitrite [9,10]. A significant intracellular source of ROS originates from mitochondria [11,12]. Additionally, oxidizing enzymes such as nicotinamide adenine dinucleotide phosphate (NADPH) oxidase, xanthine oxidase (XO), lipoxygenase (LOX), cyclooxygenase (COX), myeloperoxidase (MPO), nitric oxide synthase (NOS), and uncoupled endothelial nitric oxide synthetase (eNOS) occur in macrophages, vascular smooth muscle cells, and endothelial cells. These enzymes are involved in smooth muscle cell proliferation, the production of ROS/RNS, and LDL oxidation [7,13,14,15,16,17,18]. ROS can damage the cellular functions of biomolecules such as lipids, proteins, and carbohydrates and may be initiated, for instance, postprandially, by hyperglycemia, and/or hyperglyceridemia, resulting in lipid peroxidation and LDL oxidation [19,20]. However, any tendency towards ensuing damage is dependent not only on the site and extent of ROS generation but also whether there is any compensatory adaptive response [9].

The oxidation of lipoproteins is an initial phase in the development of atherosclerosis with deleterious and toxic effects on endothelial cells [21]. Subsequently, Ox-LDL may induce endothelial dysfunction, the expression of adhesion molecules, the migration and proliferation of smooth muscle cells, and foam cell formation [16,22]. 

Atherogenesis and atherosclerosis involve not only inflammation but also other complex processes, and this insidious and progressive disease may start before adulthood [23,24]. Atherosclerosis has three important stages heralded by a fatty streak formation, the induction of atheroma, and atherosclerotic plaques which eventually lead to atherothrombosis [25]. Lipoproteins, especially low-density lipoprotein (**LDL-C**), accumulate in the intimal layer of the arteries, and certain risk factors tend to facilitate penetration of LDL-C into the vascular intima. ROS and RNS convert the LDL-C to reactive Ox-LDL, and it remains in the vascular intima [26,27,28,29]. The activation of endothelial cells is performed by cytokines and oxidized lipids. Cytokines such as tumor necrosis factor (TNF-*α*), interleukin-1,-4, and -6 (IL-1, IL-4, IL-6) and interferon gamma (IFN-*γ*) induce the expression of leukocyte and monocyte adhesion molecules, especially vascular cell adhesion molecule-1 (VCAM), intercellular adhesion molecule-1 (ICAM) and E-selectin, on the endothelial surface [26,27,28,30]. Monocytes and T lymphocytes accumulate in the vascular wall intima mediated by these adhesion molecules [30]. Specific chemokines cause smooth muscle cell migration from media to intima and then cellular proliferation. Monocytes in the sub-endothelial space are differentiated into macrophages through chemotactic proteins, such as monocyte chemotactic protein-1 (MCP-1), macrophage colony-stimulating factor (M-CSF), and IL-8 [30,31,32]. Mononuclear phagocytosis occurs in the foam cell formation stage, and macrophages recognize and uptake Ox-LDL molecules via “scavenger” receptors (SRs). Finally, macrophages become foam cells, and the aggregation of yellow foam cells on the arterial walls leads to development of fatty streaks. The demise of foam cells occurs by programmed cell death or apoptosis during the developing atherosclerotic lesion. A necrotic core is formed as the result of apoptotic foam cell death, and this acts as a depot for cellular debris and lipids [32,33,34,35]. 

A fibrous atherosclerotic plaque cap is created during the migration of smooth muscle cells from media to intima vascular layers and the induction of extracellular matrix production in atheroma formation. The atherosclerotic plaque cap consists of collagen-rich fiber tissues, smooth muscle cells (SMC), macrophages, and T lymphocytes. Atheromatous lesions are developed by tissue macrophages and decreased blood flow in the vessels [34,35,36], and atherosclerotic plaque formation is mediated by components of the fibrous cap. Macrophages and T lymphocytes secrete metalloproteinase and TNF-α in the margins of the developing plaque in order to lyse the fibrous cap extracellular matrix and inhibit collagen synthesis in the SMC, respectively. The lysis of the extracellular matrix leads to the destruction of the fibrous cap, and the thrombogenic contents are exposed to the blood stream initiating the coagulation process, blood clot formation, the adhesion of platelets, and thrombus formation, which may completely block the arteries [37,38,39]. Figure 1 depicts the main atherosclerotic events. 

### 3.2. Antioxidants and Atherosclerosis

#### Antioxidant Defense Mechanisms 

An antioxidant is a molecule that is capable of “neutralizing” the oxidation of ROS before they react with cellular biomolecules and change their structure or function [39,40]. Antioxidant defense has two levels: 

a. Primary defense mechanism: This defense mechanism inhibits oxidative damage directly by scavenging free radicals before they can damage intracellular biomolecules. Endogenous enzymes play an important role in this step [9,41,42]. Primary defense mechanisms are summarized into four steps shown below. 

Superoxide dismutase (SOD) converts the superoxide radical generating hydrogen peroxide (H_2_O_2_): $$O_{2}^{-}+O_{2}^{-}+2H^{+}\overset{\mathrm{SOD}}{\to }H_{2}O_{2}+O_{2}$$

H_2_O_2_ is then transformed by the enzymes catalase and glutathione peroxidase (GPx) into water and molecular oxygen: $$2H_{2}O_{2}\overset{\mathrm{Catalase}}{\to }{2H}_{2}O+O_{2}$$

Glutathione peroxidase is an enzyme that catalyzes the reduction of H_2_O_2_ to water utilizing Glutathione (GSH):$$H_{2}O_{2}+2\mathrm{GSH}\overset{\mathrm{GPx}}{\to }\mathrm{GSSG}+{2H}_{2}O$$

Glutathione disulfide (GSSG) is reduced back to GSH by GSH reductase:$$\mathrm{GSSG}+\mathrm{NADPH}+H^{+}\overset{\mathrm{GSH}\;\mathrm{reductase}}{\to }2\mathrm{GSH}+{\mathrm{NADP}}^{+}$$

b. Secondary defense mechanism (chain-breaking defense): Vitamin C, vitamin E, and uric acid scavenge free radicals as a part of the secondary defense system. Additionally, nuclear enzymes which participate in DNA repair can be considered as a secondary defense system against oxidative damage caused by oxygen free radicals [9,41]. Figure 2 depicts the protective effects of different antioxidants on the stages of atherosclerosis. 

Overall, antioxidants may be categorized into two groups: Enzymatic and non-enzymatic antioxidants. Endogenous enzymatic antioxidants include superoxide dismutase (SOD), catalase (Cat), glutathione peroxidase (GPx), and thioredoxin reductase (TrxR) [9,39,43,44,45]. Endogenous non-enzymatic antioxidants include glutathione (GSH), uric acid, bilirubin, coenzyme Q (CoQ)/CoQH2) and lipoic acid [46]. 

Examples of exogenous non-enzymatic antioxidants are represented by α-tocopherol (vitamin E), ascorbic acid (vitamin C), B vitamins, carotenoids, and polyphenols. [9]. Data concerning the important enzymatic, non-enzymatic and synthetic antioxidants and their mechanisms of actions are summarized in Table 1.

### 3.3. Endogenous Enzymatic Antioxidants

#### 3.3.1. Superoxide Dismutase (SOD)

Superoxide dismutase neutralizes superoxide, thus preventing peroxynitrite formation and the reduction of transition-metal ions [10]. SOD catalyzes the superoxide anion radical to yield H_2_O_2_ and O2. It can also decrease atherosclerotic lesion size by reducing levels of F2-isoprostanes and isofurans in the aorta as well as through the inhibition of any involvement of MCP-1and VCAM-1 [23,47].

#### 3.3.2. Catalase

Catalase exists in peroxisomes and converts hydrogen peroxide (H_2_O_2_) that is formed by the dismutation of superoxide to H_2_O [10,48,49]. What is more, this enzyme also induces a decline in vascular smooth muscle cell (VSMC) proliferation [48,49].

#### 3.3.3. Glutathione Peroxidase (GPx)

GPx is a selenocysteine enzyme that reduces peroxides, especially lipid hydroperoxides, to the corresponding alcohols [10]. The most important mechanisms of GPx in the prevention and treatment of atherosclerosis stem from the inhibition of H_2_O_2_-mediated expression of MCP-1 and VCAM-1, in addition to an anti-inflammatory action [10,50].

#### 3.3.4. Thioredoxin Reductase 

The mechanism the of action of thioreductase arises from redox regulation in signaling and cell survival, increasing nitric oxide (NO) bioavailability and decreasing oxidative stress and any resultant lesions [51,52,53].

### 3.4. Endogenous Non-Enzymatic Antioxidants

Endogenous non-enzymatic antioxidants are small molecules found either intracellularly or extracellularly in a lipid or aqueous environment. Examples include glutathione, coenzyme Q, bilirubin, uric acid, and lipoic acid [23]. 

#### 3.4.1. Glutathione

Glutathione is an important small water-soluble tripeptide antioxidant present in cells [54,55], and it is a co-factor for antioxidant enzymes, such as GPx. A major action of glutathione in atherosclerosis is the modulation of the size of atherosclerotic lesions in the aortic arch by scavenging hydroxide (OH), hypochlorous acid (HOCl), and peroxynitrite (ONOO**^−^**) [54]. 

#### 3.4.2. Coenzyme Q (CoQ)

Coenzyme Q is present in cellular membranes. It is a lipophilic antioxidant with anti-inflammatory properties, and coenzyme Q10 is the main form found in humans. CoQ inhibits lipid and protein oxidation and reduces the conversion of α-tocopheroxyl radical to α-tocopherol. It is capable of scavenging peroxyl radicals, thereby improving endothelial function [56,57,58,59,60].

#### 3.4.3. Bilirubin

The key actions of bilirubin as an endogenous non-enzymatic antioxidant are via scavenging oxidants, inhibiting protein oxidation, attenuating endothelial activation/dysfunction plus SMC proliferation [61,62,63,64,65], inhibiting in vivo leukocyte adhesion to endothelial cells [66], and the phosphorylation of retinoblastoma tumor suppressor protein in addition to p38 mitogen-activated protein kinase (p38 MAPK) [64,67].

#### 3.4.4. Uric Acid

Uric acid is the end-product of purine catabolism. There are two immediate sequential precursors—hypoxanthine followed by xanthine—the conversions of which are both catalyzed by xanthine oxidase. Mechanistically, uric acid increases cytokine production, scavenges OH as well as HOCl, and incites inflammatory responses, SMC proliferation, endothelial dysfunction, and plaque instability [68]. 

#### 3.4.5. Lipoic Acid

Lipoic acid is synthesized by mitochondria. It is a cofactor for mitochondrial α-ketoacid dehydrogenases (e.g., the pyruvate dehydrogenase complex), and it inhibits atherosclerotic lesion development [69]. It is also a scavenger of ONOO-, HOCl, and peroxyl radicals. Other key actions of lipoic acid comprise an attenuation of endothelial dysfunction, a decrease in inflammatory markers, and an increase in endothelial nitric oxide synthase (eNOS) activity [70,71,72]. 

### 3.5. Exogenous Non-Enzymatic Antioxidants (Natural Antioxidants) 

#### 3.5.1. Vitamin E

Vitamin E is the most comprehensively studied lipid soluble antioxidant in humans. It consists of eight isomeric compounds (α-, β-, γ-, and δ- tocopherol; and α-, β-, γ-, and δ-tocotrienol) [9,40]. Cooking oils, egg yolk, butter, green leafy vegetables, and some fruit (kiwi fruit, pumpkins, mangoes, papayas, and tomatoes) are rich sources of vitamin E [9].

In several animal models, vitamin E has preventative effects against atherosclerosis by: Scavenging free radicals in VSMC, diminishing the oxidation of LDL by the inhibition of Cluster Differentiating 36 (CD36) and Scavenger receptor class B type I (SR-BI) expression in VSMC, reducing VSMC proliferation via the inhibition of protein kinase C (PKC), preventing foam cell formation, lessening the secretion of cytokines and extracellular matrix in VSMC, preventing mononuclear cell infiltration, lessening inflammation, curtailing the destabilization of fibrous plaque, inhibiting the apoptosis of VSMC, modulating signal transduction and gene expression in VSMC, increasing the expression of connective tissue growth factor (CTGF) in VSMC (cell lines), preventing endothelial dysfunction related to cholesterol, modulating endothelial cells and the expression of adhesion molecules such as VCAM-1 and ICAM-1 on endothelial cells, preventing lysophosphatidylcholine (LPC)-induced endothelial dysfunction and the preservation of endothelial NO release, modulating monocytes, macrophages, T cells and mast cells, enhancing the expression of cytosolic phospholipase A2 (PLA2), cyclooxygenase, and vasodilating prostacyclin (PGI2) in endothelial cells, inhibiting thrombin formation, and reducing leukotriene synthesis [73,74,75,76,77,78,79,80,81,82].

In several clinical studies, vitamin E revealed contrasting findings. In a study examining the effect of 50 mg·day^−1^ synthetic vitamin E in a population with coronary heart disease, the results showed no effect on major cardiovascular events [83]. Another study showed 300 mg day^−1^ synthetic vitamin E had no effect on cardiovascular disease, including the rate of non-fatal myocardial infarction in patients with previous myocardial infarction [84]. Additionally, vitamin E did not significantly decrease the incidence of cardiovascular disease such as stroke [85]. In the Cambridge heart antioxidant study (CHAOS), vitamin E supplementation failed to have an impact on cardiovascular outcomes in patients at high risk of cardiovascular events [86].

#### 3.5.2. Vitamin C

Vitamin C (ascorbate) is a water-soluble and ubiquitous antioxidant [7,23] with an ability to scavenge peroxyl radicals and HOCL [23,40], thus providing stability to the cell membrane. Fruit and vegetables, particularly citrus fruit, kiwi, cantaloupe, mango, strawberries, and peppers are rich sources of vitamin C [9].

It has various functions including: The improvement of nitric oxide-dependent vasomotor function [87], the enhancement of NOS activity (NO production) and the consequent augmentation of NO bioavailability, the improvement of endothelial dysfunction and vasodilation, the inhibition of cyclooxygenase, the diminishing of cell–cell adhesion [88], and the reduction of the chain-carrying α-tocopheroxyl radical to inhibit LDL peroxidation [89]. It also recycles other endogenous antioxidants, such as vitamin E [90]; discourages leukocyte aggregation and adhesion to the endothelium [9]; and scavenges ROS such as superoxide, hydroxyl radicals, peroxyl radicals, and many non-radicals, such as nitrosating agents and hydrochlorous acid [9]. 

A number of small-scale clinical studies have evaluated the effect of vitamin C on vascular health. The British Regional Heart Study demonstrated an inverse relation between plasma vitamin C concentration and endothelial dysfunction in men with no history of cardiovascular disease or diabetes [91]. Additionally, the European prospective investigation into cancer and nutrition (EPIC) Norfolk study showed the same results as the British Regional Heart Study in both men and women [92]. A large-scale study conducted over 20 years found that diets rich in vitamin C had no significant association with coronary heart disease [93].

#### 3.5.3. B Vitamins 

B vitamins have a fundamental role in the metabolism of essential amino acids, with a specific influence on homocysteine and the antioxidant, glutathione [9]. Other significant activities of B vitamins entail scavenging hydroxyl and lipid peroxyl radicals, improving endothelial function, and ameliorating the coupling of endothelial NO synthase through the essential cofactor, tetrahydrobiopterin [94,95].

In a clinical study intake of folate, hydroxocobalamin, and pyridoxine, supplements for eight weeks decreased serum homocysteine to a normal range in patients with venous thrombosis [96]. The vitamin intervention for stroke prevention (VISP) randomized controlled trial study demonstrated no significant effect of folate, hydroxocobalamin, and pyridoxine supplementation in decreasing incidence of coronary events or cardiovascular death [97]. The Cochrane systematic review reported no evidence to prevent cardiovascular events by using B vitamins [98]. 

#### 3.5.4. Vitamin A and Carotenoids

Carotenoids are a large group of lipid soluble, colorful substances (yellow, orange, and red) such as α-carotene, β-carotene, β-cryptoxanthine, luteine and lycopene which occur extensively in fruit and vegetables [9]. 

They scavenge free radicals and prevent LDL peroxidation. β-Carotene can decrease plasma cholesterol levels by inhibiting HMG-CoA reductase (3-hydroxy-3-methyl-glutaryl-coenzyme A reductase). In addition, carotenoids are capable of increasing macrophage LDL receptor activity and reducing circulating LDL, inflammation, oxidative stress, and endothelial dysfunction [99,100]. 

A clinical study suggested an inverse relationship between the intake of β-carotene or retinol and risk of cardiovascular disease [101]. The US Preventative Task Force does not suggest β-carotene for the prevention of cardiovascular disease [102]. The Cochrane review on antioxidant consumption indicated that β-carotene and vitamin A significantly increase all-cause mortality [103]. 

#### 3.5.5. Polyphenols

Polyphenols are the most abundant natural antioxidants possessing variable phenolic structures. They are found in fruit (especially apples), grains, vegetables, cereals, olive oil, dry legumes, chocolate and beverages, such as tea, coffee and wine [104]. These compounds are divided into several classes: Flavonoids, phenolic acids (e.g., caffeic acid and gallic acid), stilbenes (e.g., resveratrol), and lignans (e.g., secoisolariciresinol) [9]. Flavonoids, which are a major class of polyphenols, are subclassified as flavonols (e.g., quercetin), flavones (e.g., apigenin, luteolin), flavanones (naringenin, hesperetin), flavan-3-ols (catechins and their oligomers: Proanthocyanidins), isoflavones (e.g., genstein), and anthocyanins (e.g., delphinidin, cyanidin) [9,104].

The mechanistic effects of polyphenols involve: Suppressing ROS formation, scavenging ROS (both radical and non-radical oxygen), increasing the expression level of eNOS and the generation of NO or reducing NO oxidation by enhancing the intracellular free calcium concentration and by activating estrogen receptors in endothelial cells (ECs), blocking the action of xanthine oxidase and protein kinase C to prevent the production of the superoxide radical, and the protection of vascular endothelial cells and NO from oxidation. They also decrease redox-sensitive gene activation, preventing the expression of two major pro-angiogenic factors (vascular endothelial growth factor (VEGF) and matrix metalloproteinase-2 (MMP-2)) in smooth muscle cells, increase the production of major vasodilatory factors (NO, endothelium-derived hyperpolarizing factor (EDHF) as well as prostacyclin), inhibit angiogenesis (cell migration and proliferation of blood vessels), and also reduce platelet aggregation and hypertension [105,106,107,108,109,110].

Another important polyphenol that has received much attention is resveratrol (3, 5, 4′-trihydroxystilbene), a stilbene polyphenol, which occurs in grapes, red wine and *Polygonum cuspidatum*. Resveratrol has established antioxidant properties which include the inhibition of lipid oxidation, the regulation of vasodilator and vasoconstrictor production, the inhibition of platelet aggregation, and the inhibition of the transcription factors NF-κB (Nuclear Factor kappaB) and AP-1 (Activator Protein 1) through an interaction with upstream signaling pathways and/or by decreasing pro-inflammatory mediators (TNF-α, IL) [104,111]. 

Clinical studies such as the Zupthen Elderly study showed a significant inverse association between flavonoid intake and coronary heart disease after 5 years of consumption [112,113]. In addition, the Rotterdam study revealed a significant inverse relationship between total flavonoid intake from the diet with myocardial infarction incidence [114]. The consumption of cocoa or chocolate is inversely associated with carotid atherosclerosis [115].

### 3.6. Synthetic Antioxidants: Probucol and Related Phenols

Probucol (2,6-di-tert-butyl-4-({2-[3,5-di-tert-butyl-4-hydroxyphenyl)sulfanyl) propan-2-yl} sulfanyl)phenol) is a phenolic lipid-soluble antioxidant [7]. Its activities related to any antiatherosclerotic effectiveness consist of anti-inflammatory activity, the augmentation of endothelial function and repair, lessening oxidant production in vessel walls, attenuating atherosclerosis through the inhibition of LDL oxidation by blocking the production of oxidizing intermediates, inducing heme oxygenase-1 (HO-1) in arterial cells, inhibiting vasomotor dysfunction and fatty streak formation [87], reducing restenosis [7,116,117], inhibiting smooth muscle cell proliferation and adhesion molecule expression on endothelial cells, and promoting endothelium-dependent vasomotion [7].

Another important lipophilic and synthetic antioxidant is BO-653n (2, 3-dihydro-5-hydroxy-2, 2-dipentyl-4,6-di-tert-butylbenzofuran) (an analog of α-tocopherol) which inhibits the formation of atherosclerotic lesions [26], reduces α- tocopheroxyl radical and inhibits LDL oxidation in the intimal area [23,116,118,119].

### 3.7. Effective Medicinal Plants on Atherosclerosis

Medicinal plants can be employed safely to prevent and treat atherosclerosis, mainly because of their tendency to produce fewer adverse effects [8]. They commonly possess antioxidant activity, representing a key underlying mechanism often linked to their phenolic constituents [120]—though their efficacy may also derive from combinations of other properties. The most effective medicinal plants used to prevent or treat atherosclerosis, along with their known mechanisms of action other than antioxidant activity, are listed in Table 2. 

### 3.8. Reasons for Failing Antioxidant Strategies Related to Atherosclerosis in Humans

Several antioxidants have been tested with positive effects in different animal models for the potential treatment of atherosclerosis. However, studies in humans are either limited or have not disclosed positive effects. The reasons accounting for the failure of traditional antioxidant therapy in humans may be attributed to the following factors: Antioxidants should be utilized in the long term so that beneficial effects may be allowed an adequate period to emerge.Antioxidant treatment should ideally be instigated before full disease onset.Antioxidants (e.g., vitamin E) may ultimately lose beneficial effects through oxidation.The oxidant theory of atherogenesis is essentially a deficient and incomplete theory and does not incorporate effects of other pathways in atherogenesis.It is evident that the antioxidants that pass through the mitochondrial membrane, thus modifying mitochondrial oxidation, have superior effectiveness compared to traditional antioxidants.Combination antioxidant therapies may prove to be more effective overall because they may exploit any additional constituent mechanistic properties [16].

Diet plays an important role in the prevention of atherosclerosis and other cardiovascular diseases. In fact, the lifestyle habits, nutritional quality, and acquired eating patterns are effective on the risk of atherosclerosis [158,159]. The risks of atherosclerosis are decreased through an appropriate balance of nutrients. In fact, the balance of calorie intake and physical activity to keep a healthy body weight is very important. In this regard, the following diet and lifestyle have been recommended:—Intake a diet rich in vegetables, fruits (300 g/day of fruit or 400 g/day of vegetable consumption); whole-grain cereals (women 75 g/day, men 90 g/day); extra-virgin oil (≥4 tbsp/day); nuts (3e7 servings/week); a moderate consumption of fish and poultry (≥3 servings/week); a low intake of dairy products, red meat and sweets; and a moderate consumption of red wine for usual drinkers (≥7 glasses/week, average dietary fiber intake was higher than 30 g/day).—Consumption of foods and beverages with little salt and added sugars.—Reduction of trans-fat to <1% of energy, saturated fat to <7% of energy, and cholesterol to <300 mg/day by consuming lean meats and vegetable alternatives.—Low level of hydrogenated fats.—Intake of 2–3 g/day of plant sterol/stanol esters to reduce LDL-C.—Supplementation with >500 mg vitamin C/day.—Consumption of high doses of resveratrol (≥150–1000 mg/day).—Consumption of adequate micronutrients such as potassium (10 gr), magnesium (500 mg), and zinc (45 mg).—Consumption of 25 g of soy protein and 15 g of soluble fiber daily for two months,—Intake of adequate vitamins such as vitamin E (400 to 1200 IU/day), vitamin C (≥250 mg/day) and vitamin D (≥30 ng/mL).—Intake of 50 g of dark chocolate, 100 mg of flavanols, and 500–1000 mg/day quercetin.—Drinking ≥3 cups daily of tea (black or preferably green tea).—Supplementation with standardized colored plants such as the maqui berry (162 mg anthocyanins) [158,159,160].

## 4. Conclusions

Atherosclerosis is a major cause of morbidity and mortality in the developed world. Due to the factor that ROS and the generation of oxidized LDL are leading contributors to the progression of atherosclerosis, dietary supplements and antioxidants with low adverse effects may well represent a good therapeutic strategy to prevent the progression of the disease. 

Natural and synthetic antioxidants facilitate atherosclerosis treatment through a variety of mechanisms, including the inhibition of LDL oxidation, the reduction of generated reactive oxygen species, the inhibition of cytokine secretion, the prevention of atherosclerotic plaque formation and platelet aggregation, the prevention of mononuclear cell infiltration, the improvement of endothelial dysfunction and vasodilation, the promotion of NO bioavailability, the modulation of the expression of adhesion molecules such as VCAM-1 and ICAM-1 on endothelial cells, and the suppression of foam cell formation.

It is not clear which of these different mechanisms of antioxidants action is more effective, but it seems that the use of multiple antioxidants is more effective target for antioxidant therapy.

## Figures and Tables

**Figure 1 biomolecules-09-00301-f001:**
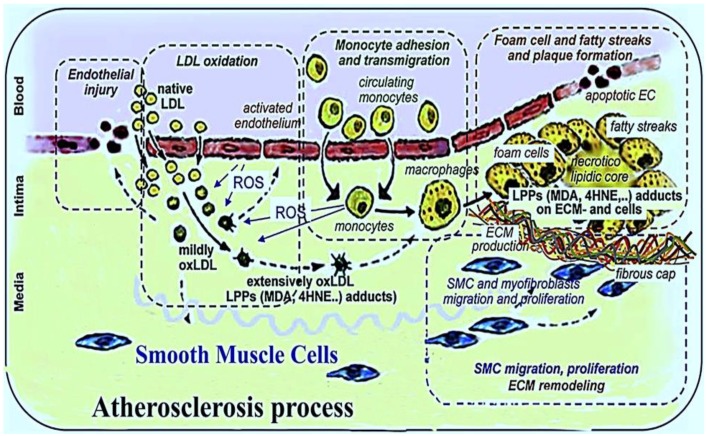
The main atherosclerotic events [39].

**Figure 2 biomolecules-09-00301-f002:**
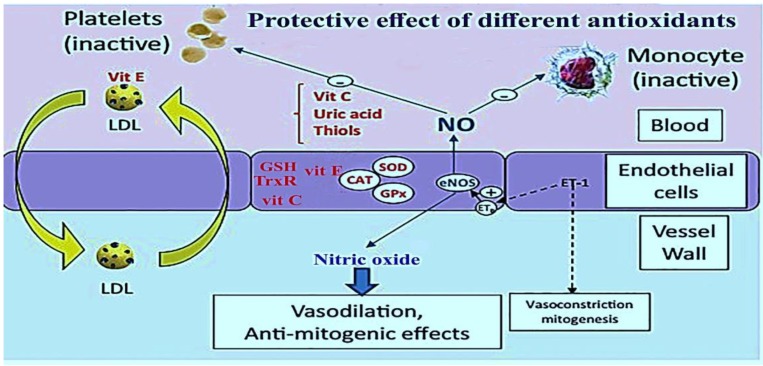
The protective effects of antioxidants on the stages of atherosclerosis [9].

**Table 1 biomolecules-09-00301-t001:** Important enzymatic, non-enzymatic, and synthetic antioxidants.

Antioxidants	Type of Antioxidants	Type of Antioxidants	Action Mechanism	Reference
Enzymatic	endogenous	Superoxide dismutase	Preventing peroxynitrite formation, reducing levels of F2-isoprostanes and isofurans in the aorta and reduction of transition-metal ions	[23,47]
		Catalase	Reducing vascular smooth muscle cell (VSMC) proliferation	[48,49]
		Glutathione peroxidase	Inhibition of H_2_O_2_-mediated expression of MCP-1 and VCAM-1	[10,50]
		Thioredoxin reductase	Increasing NO bioavailability and decreasing oxidative stress	[51,52,53]
Non-enzymatic	endogenous	Glutathione	Modulation of the size of atherosclerotic lesions in the aortic arch	[54]
		Uric acid	Increasing cytokine production, scavenging OH, plaque instability, attenuating endothelial activation and SMC proliferation	[68]
		Bilirubin	Scavenging oxidants, inhibiting protein oxidation, attenuating endothelial activation/dysfunction and SMC proliferation	[61,62,63,64,65]
		Coenzyme Q	Inhibiting lipid and protein oxidation, scavenging peroxyl radicals and improving endothelial function	[56,57,58,59,60]
		Lipoic acid	Inhibition atherosclerotic lesion development, increasing in eNOS activity	[69,70,71,72]
	exogenous	Vitamin E	Preventing foam cell formation and endothelial dysfunction, scavenging free radicals, diminishing the oxidation of LDL and modulating endothelial cells	[73,74,75,76,77,78,79,80,81,82]
		Vitamin C	Enhancement of NOS activity, inhibition of cyclooxygenase, diminishing cell–cell adhesion, improvement of endothelial dysfunction and vasodilation	[87,88,89]
		B vitamins	Scavenging hydroxyl and lipid peroxyl radicals, improving endothelial function	[94,95]
		Vitamin A and Carotenoids	Prevention LDL peroxidation, reducing inflammation, oxidative stress, and endothelial dysfunction	[99,100]
		Polyphenols	Suppressing ROS formation, increasing the expression level of eNOS, inhibiting angiogenesis, reducing platelet aggregation and hypertension	[105,106,107,108,109,110]
Synthetic	Synthetic	Probucol	Augmentation of endothelial function and repair, inducing heme oxygenase-1 (HO-1) in arterial cells, inhibiting vasomotor dysfunction and fatty streak formation, inducing heme oxygenase-1 (HO-1) in arterial cells	[87,116,117]
		BO-653n	Reduces α-tocopheroxyl radical and inhibits LDL oxidation in the intimal area	[116,117,118,119]

**Table 2 biomolecules-09-00301-t002:** Non-antioxidant potential mechanisms in medicinal plants with antioxidant effect *.

Action Mechanism	Medicinal Plants	Reference
Endothelial protective activity	*Rhizoma polygonum*	[121]
	*Salvia miltiorrhiza*	[122]
	*Buddleja officinalis*	[123]
	*Tribulus terrestris*	[124]
	*Panax notoginseng*	[125]
	*Ginkgo biloba*	[126]
	*Curcuma longa*	[8]
	*Magnolia officinalis*	[127]
Lowering blood lipid levels and regulation of inflammatory processes	*Ocimum basilicum*	[128]
	*Tribulus terrestris*	[124]
	*Artemisia aucheri*	[129]
	*Terminalia arjuna*	[130]
	*Cynanchum wilfordii*	[131]
	*Celastrus orbiculatus*	[132]
Suppression of foam cell formation	*Arisaema tortuosum*	[133]
	*Rhododendron dauricum*	[134]
	*Celastrus orbiculatus*	[132]
	*Terminalia arjuna*	[130]
	*Chlorophytum borivilianum*	[135]
	*Buddleja officinalis*	[136]
	*Lycium barbarum*	[137]
	*Scutellaria baicalensis*	[138]
	*Rheum rhabarbarum*	[8]
	*Glossogyne tenuifolia*	[139]
	*Paeonia lactiflora*	[8]
	*Achyrocline satureoides*	[140]
	*Cassia tora*	[141]
	*Gynostemma pentaphyllum*	[8]
	*Artemisia scoparia*	[142]
	*Panax pseudoginseng*	[143]
	*Camellia sinensis*	[144]
	*Mellilotus Officinalis*	[120]
	*Zingiber officinalis*	[145]
Suppression of both monocyte migration/activation plus foam cell formation	*Prunella vulgaris*	[146]
	*Panax notoginseng*	[125]
	*Phyllanthus emblica*	[147]
Suppression of vascular smooth muscle cell (VSMC) migration and proliferation plus suppression of foam cell formation.	*Gleditsia sinensis*	[148]
	*Nelumbo nucifera*	[149]
	*Hibiscus sabdariffa L.*	[150]
	*Astragalus membranaceus*	[151]
Inhibition of platelet aggregation, coagulation and antiplatelet activity	*Allium sativum*	[152,153]
	*Aronia melanocarpa*	[154]
	*Coptis Chinensis*	[155]
Anti-lipid effects	*Nigella sativa*	[156]
	*Cynara scolymus*	[157]

* These plants all possess antioxidant activity which may inevitably contribute significantly to their overall effectiveness.

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
