# Peer review of "Antioxidants and Atherosclerosis: Mechanistic Aspects"

_biomolecules, 2019, doi:10.3390/biom9080301_

Round 1

Reviewer 1 Report

The manuscript by Malekmohammad et al describes the role of antioxidants in atherosclerosis, with specific emphasis in the mechanism of action. This is a comprehensive and well-written review, providing descriptions on numerous antioxidant activities. However, it would benefit from some rearrangements that would make it more focused and easier for the reader to follow.

1) A Figure depicting the main atherosclerotic events described in the text would be helpful to the reader

2) Is there any evidence regarding the stages of atherosclerosis that are targeted by the antioxidants described? If so, it would be helpful to also include in a figure

3) While it is important to include information on all antioxidants with different modes of action (enzymatic, non-enzymatic, natural, synthetic), the authors could consider including information for only a few examples per category, preferably selecting ones with more research evidence available. Key points on mode of action of all antioxidants could be provided in a table format. In this way, information provided to the reader will be more focused and easier to follow.

4) Regarding the medicinal plants, please include a brief description on their antioxidant properties. The table included provides their mechanism of action other than antioxidant activity. Which of these medicinal plants exert antioxidant activities? Is there any evidence on the underlying mechanism of this property?

5) A brief description of the main findings from the use of antioxidants in clinical studies would be useful. What were the clinical parameters examined, what drugs were used and how did these fail to protect from atherosclerosis?

6) Since antioxidants have so far failed to produce positive outcomes in atherosclerosis patients, how do the authors envision their future as therapeutic strategies? Would other strategies, (eg lipid lowering or apo-mimetics), that also exert some antioxidant properties hold promise as future therapies? 

7) Among the different mechanisms of antioxidants action that are described in the conclusion section, which do the authors believe would be the most promising to target for future therapies?

Minor

1) Please ensure that font size is the same throughout the text  

Author Response

Dear reviewer,

The manuscript was revised based on the respective your comments as follows. The changes were highlighted on the manuscript.

Reviewer 1

The manuscript by Malekmohammad et al describes the role of antioxidants in atherosclerosis, with specific emphasis in the mechanism of action. This is a comprehensive and well-written review, providing descriptions on numerous antioxidant activities. However, it would benefit from some rearrangements that would make it more focused and easier for the reader to follow.

Thanks a lot.

1) A Figure depicting the main atherosclerotic events described in the text would be helpful to the reader.

A figure was added in the manuscript.

2) Is there any evidence regarding the stages of atherosclerosis that are targeted by the antioxidants described? If so, it would be helpful to also include in a figure

A figure was added in the manuscript.

3) While it is important to include information on all antioxidants with different modes of action (enzymatic, non-enzymatic, natural, synthetic), the authors could consider including information for only a few examples per category, preferably selecting ones with more research evidence available. Key points on mode of action of all antioxidants could be provided in a table format. In this way, information provided to the reader will be more focused and easier to follow.

A table about information on all antioxidants with different modes of action was added in the manuscript (table 1).

4) Regarding the medicinal plants, please include a brief description on their antioxidant properties. The table included provides their mechanism of action other than antioxidant activity. Which of these medicinal plants exert antioxidant activities? Is there any evidence on the underlying mechanism of this property?

The provided plants all possess antioxidant activity which may contribute significantly to their overall effectiveness. Therefore, the title of the table was changed and corrected in the manuscript. (Title of the table 2. Non-antioxidant potential mechanisms in medicinal plants with antioxidant effects).

5) A brief description of the main findings from the use of antioxidants in clinical studies would be useful. What were the clinical parameters examined, what drugs were used and how did these fail to protect from atherosclerosis?

Clinical studies about the use of antioxidants were added in the manuscript.

6) Since antioxidants have so far failed to produce positive outcomes in atherosclerosis patients, how do the authors envision their future as therapeutic strategies? Would other strategies, (eg lipid lowering or apo-mimetics), that also exert some antioxidant properties hold promise as future therapies? 

The use of antioxidants as a therapeutic strategy should be identified with large– scale studies, and if they were not effective, it should be focused on strategies such as lipid lowering or apo-mimetics.

7) Among the different mechanisms of antioxidants action that are described in the conclusion section, which do the authors believe would be the most promising to target for future therapies?

It is not clear which of these mechanisms of antioxidants actions are more effective, but it seems that the use of multiple antioxidants is more effective target for therapies. It was added in the conclusion section of manuscript.

Minor

1) Please ensure that font size is the same throughout the text  

It was checked

Best Wishes,

Prof. Mahmoud Rafieian-Kopaei

Reviewer 2 Report

This paper omits many  treatments that must be discussed.  It then lists a lot of herbs without any discussion of their validity.   It does not review surrogate endpoints or hard endpoints for CHD such as carotid IMT, plaque,  CAC, MI, CHD etc.  Here are only a few of the omissions

1.  Diet/ and dietary nitrate : 0.1 mmol/kg of body weight /day. 10

          servings of fruits (4)(berries) and vegetables (6) with dark green

          leafy vegetables.  DASH  2 and Mediterranean diets. Caloric  

          restriction (12.5/12.5 EE with overnight fast. 30 % protein, 30   

          % MUFA and omega 3 FA with limited SFA and no trans fat,

          minimal refined CHO(50 grams), more complex CHO (40%).

          Consume smaller meals more frequently with antioxidants/meal

          Minimal caffeine depending on CYP 1A2 status.

    2.  Vitamin C sustained release : 250-500 mg bid.

    3.  Vitamin K 2 MK 7   200-1000 mcg per day

    4.  Polyphenols:  20 grams dark chocolate (>70%), EGCG 500 mg bid or green tea 32 oz/day ( decaffeinated), 6 ounces red wine.

    5.  Quercetin 500-1000 mg/day.

    6.  Curcumin 500 mg-1000 mg bid.

    7.  2 gram sodium, 10 gram potassium, 1000 mg magnesium /day

    8.  500 mg beetroot juice: 45 mmol/L or 2.79 g/L inorganic nitrate/day.

    9.  Pomegranate seeds: 1/2 cup per day or juice 6 ounces/day.

.   BH4  2mg/kg/day with 5 methyl folate 1000 -5000ug   

            per day with B complex vitamins.

     10.  R-lipoic acid (RLA) 100 mg per day with biotin 5000

            ug/day for  GSH (glutathione)  and acetyl –L-carnitine 1000 mg/d 

           (mitochondrial function)

     11.  NAC(n-acetyl cysteine) 500 mg bid for GSH (glutathione) etc.

     12.  Whey protein 30-40 grams per day for GSH (glutathione)

     13.  Niacinamide 500-1000 mg bid for GSH ( glutathione) etc.

     14.  MSM  500 mg bid

     15.  Branched chain amino acids ( leucine , valine, isoleucine  4:1:1 ratio) 

            5000 mg/d

     16.  D-Ribose 5 grams tid and nicotinomide riboside one per day

     17.  Phosphatidyl Serine   300-600 mg bid

     17.  Trans-resveratrol 250 mg per day with grape seed 

            extract 500 mg bid

     18 .  Balanced omega 3 FA (DHA, EPA, GLA with gamma delta

             tocopherols: 2- 5 grams per day

Plant      sterols   2.5 grams per day and      sterolins.

Reishi  and Shiitake mushrooms: one serving per      day.

Vitamin      D3  to level of 60 ng/ml.

AGED      garlic (Kyolic)  CV formulation: 600      mg bid.

Co      enzyme Q 10 :100 mg per day to level of 3 ug/dl and PPQ 20 mg/d

Lycopene:      20 mg per day ( supplement, tomato, pink grapefruit, watermelon etc.).

Carnosine      500 mg bid

Berberine       500-1000 mg per day

High      quality varied multivitamin and fruit and vegetable extracts

Probiotics:  50 billion CFU per day

This paper needs MAJOR revision before it can be accepted

Author Response

Dear reviewer,

The manuscript was revised based on the respective your comments as follows. The changes were highlighted on the manuscript.

Reviewer 2

Comments and Suggestions for Authors

This paper omits many treatments that must be discussed.  It then lists a lot of herbs without any discussion of their validity. It does not review surrogate endpoints or hard endpoints for CHD such as carotid IMT, plaque, CAC, MI, CHD etc.  Here are only a few of the omissions

1.  Diet/ and dietary nitrate : 0.1 mmol/kg of body weight /day. 10

          servings of fruits (4)(berries) and vegetables (6) with dark green

          leafy vegetables.  DASH  2 and Mediterranean diets. Caloric  

          restriction (12.5/12.5 EE with overnight fast. 30 % protein, 30   

          % MUFA and omega 3 FA with limited SFA and no trans fat,

          minimal refined CHO(50 grams), more complex CHO (40%).

          Consume smaller meals more frequently with antioxidants/meal

          Minimal caffeine depending on CYP 1A2 status.

    2.  Vitamin C sustained release : 250-500 mg bid.

    3.  Vitamin K 2 MK 7   200-1000 mcg per day

    4.  Polyphenols:  20 grams dark chocolate (>70%), EGCG 500 mg bid or green tea 32 oz/day ( decaffeinated), 6 ounces red wine.

    5.  Quercetin 500-1000 mg/day.

    6.  Curcumin 500 mg-1000 mg bid.

    7.  2 gram sodium, 10 gram potassium, 1000 mg magnesium /day

    8.  500 mg beetroot juice: 45 mmol/L or 2.79 g/L inorganic nitrate/day.

    9.  Pomegranate seeds: 1/2 cup per day or juice 6 ounces/day.

.   BH4  2mg/kg/day with 5 methyl folate 1000 -5000ug   

            per day with B complex vitamins.

     10.  R-lipoic acid (RLA) 100 mg per day with biotin 5000

            ug/day for  GSH (glutathione)  and acetyl –L-carnitine 1000 mg/d 

           (mitochondrial function)

     11.  NAC(n-acetyl cysteine) 500 mg bid for GSH (glutathione) etc.

     12.  Whey protein 30-40 grams per day for GSH (glutathione)

     13.  Niacinamide 500-1000 mg bid for GSH ( glutathione) etc.

     14.  MSM  500 mg bid

     15.  Branched chain amino acids ( leucine , valine, isoleucine  4:1:1 ratio) 

            5000 mg/d

     16.  D-Ribose 5 grams tid and nicotinomide riboside one per day

     17.  Phosphatidyl Serine   300-600 mg bid

     17.  Trans-resveratrol 250 mg per day with grape seed 

            extract 500 mg bid

     18 .  Balanced omega 3 FA (DHA, EPA, GLA with gamma delta

             tocopherols: 2- 5 grams per day

Plant      sterols   2.5 grams per day and      sterolins.

Reishi  and Shiitake mushrooms: one serving per      day.

Vitamin      D3  to level of 60 ng/ml.

AGED      garlic (Kyolic)  CV formulation: 600      mg bid.

Co      enzyme Q 10 :100 mg per day to level of 3 ug/dl and PPQ 20 mg/d

Lycopene:      20 mg per day ( supplement, tomato, pink grapefruit, watermelon etc.).

Carnosine      500 mg bid

Berberine       500-1000 mg per day

High      quality varied multivitamin and fruit and vegetable extracts

Probiotics:  50 billion CFU per day

Diet, lifestyle recommendations and balance of nutrients were added in the manuscript.

Best Wishes,

Prof. Mahmoud Rafieian-Kopaei

Round 2
